# ZNF768 Expression Associates with High Proliferative Clinicopathological Features in Lung Adenocarcinoma

**DOI:** 10.3390/cancers13164136

**Published:** 2021-08-17

**Authors:** Audrey Poirier, Andréanne Gagné, Philippe Laflamme, Meagan Marcoux, Michèle Orain, Sophie Plante, David Joubert, Philippe Joubert, Mathieu Laplante

**Affiliations:** 1Centre de Recherche de l’Institut Universitaire de Cardiologie et de Pneumologie de Québec (CRIUCPQ), Faculté de Médecine, Université Laval, 2725 Chemin Ste-Foy, Québec, QC G1V 4G5, Canada; audrey.poirier.5@ulaval.ca (A.P.); andreanne.gagne.4@ulaval.ca (A.G.); philippe.laflamme.2@ulaval.ca (P.L.); meagan.marcoux.1@ulaval.ca (M.M.); michele.orain@criucpq.ulaval.ca (M.O.); sophie.plante.1@ulaval.ca (S.P.); philippe.joubert@criucpq.ulaval.ca (P.J.); 2Centre de Recherche sur le Cancer de l’Université Laval, Université Laval, 9 Rue McMahon, Québec, QC G1R 3S3, Canada; 3Faculté des Sciences Sociales, Université d’Ottawa, 125 University Private, Ottawa, ON K1N 6N5, Canada; david.joubert@uottawa.ca

**Keywords:** lung adenocarcinoma, ZNF768, cell proliferation

## Abstract

**Simple Summary:**

Zinc-finger protein 768 (ZNF768) is a transcription factor that was recently shown to promote proliferation and repress senescence downstream of growth factor signaling in vitro. This protein was found to be overexpressed in a small cohort of lung adenocarcinoma (LUAD) compared to normal tissue, but its clinical value in this cancer remains unknown. The aim of this study was to determine whether ZNF768 associates with clinicopathological features in LUAD. We found that ZNF768 levels are often elevated in LUAD and that ZNF768 protein levels positively correlate with Ki-67 and other proliferative clinicopathological features in this cancer. Supporting a role for ZNF768 in promoting proliferation, ZNF768 depletion severely impairs proliferation in several lung cancer cell lines. Our study, that extends previous in vitro data, provides the first clinical observations supporting a possible role for ZNF768 in promoting cancer cell proliferation and tumor development in humans.

**Abstract:**

Lung adenocarcinoma (LUAD) is the most common type of lung cancer and a leading cause of cancer-related deaths worldwide. Despite important recent advances, the prognosis for LUAD patients is still unfavourable, with a 5 year-survival rate close to 15%. Improving the characterization of lung tumors is important to develop alternative options for the diagnosis and the treatment of this disease. Zinc-finger protein 768 (ZNF768) is a transcription factor that was recently shown to promote proliferation and repress senescence downstream of growth factor signaling. Although ZNF768 protein levels were found to be elevated in LUAD compared to normal lung tissue, it is currently unknown whether ZNF768 expression associates with clinicopathological features in LUAD. Here, using tissue microarrays of clinical LUAD surgical specimens collected from 364 patients, we observed that high levels of ZNF768 is a common characteristic of LUAD. We show that ZNF768 protein levels correlate with high proliferative features in LUAD, including the mitotic score and Ki-67 expression. Supporting a role for ZNF768 in promoting proliferation, we report that ZNF768 depletion severely impairs proliferation in several lung cancer cell lines in vitro. A marked decrease in the expression of key proliferative genes was observed in cancer cell lines depleted from ZNF768. Altogether, our findings support a role for ZNF768 in promoting proliferation of LUAD.

## 1. Introduction

Lung cancer is amongst the most common cancer worldwide and is the leading cause of cancer-related deaths, accounting for 1.76 million deaths annually in men and women [1]. A large portion of patients are diagnosed with an advanced stage of the disease and are left with limited treatment options [2,3,4]. Recent advances in the management of lung cancer with targeted therapies and immunotherapy have greatly improved the survival rate of this disease, but despite these promising new treatments nearly all patients will develop resistance [5,6,7].

Lung cancer is classified into two main histological groups based on their clinical behavior, small cell lung carcinoma (SCLC) and non-SCLC (NSCLC), with NSCLC representing the majority of the cases. This group can further be divided into adenocarcinoma, squamous cell carcinoma and large cell carcinoma. Lung adenocarcinoma (LUAD) is the most common type of NSCLC and constitutes almost half of all lung cancer cases [8,9]. This subtype also shows a higher incidence in non-smokers and females [8,10]. Multiple genetic alterations are associated with LUAD. Based on the most recent reports of the Cancer Genome Atlas Research Network on LUAD, the driver mutations associated with LUAD are concentrated in some key pathways and processes [11,12]. These include activation of the receptor tyrosine kinase (RTK)/RAS/RAF (76%, e.g., *EGFR*, *K-RAS*, *MET*) and phosphoinositide-3-kinase (PI3K)/mechanistic target of rapamycin (mTOR) (25%, e.g., *STK11*, *PIK3CA*) pathways, inactivation of the p53 pathway (63%, e.g., *TP53*, *MDM2*) and repression of key cell cycle regulators (64%, e.g., *CDKN2A*).

Zinc-finger protein 768 (ZNF768) is a transcription factor that has been shown to greatly impact the proliferation and the survival of cancer cells in vitro [13,14]. We recently identified RAS as a novel upstream regulator of ZNF768 [13]. More specifically, ZNF768 is phosphorylated and destabilized in response to RAS activation via the mitogen-activated protein kinases (MAPK) and the PI3K/mTORC2 pathways. Supporting its role in promoting cell proliferation, we found that ZNF768 depletion impairs proliferation and induces senescence by altering the expression of major cell cycle effectors and p53 targets. Notably, we found that ZNF768 overexpression allows cells to bypass RAS-induced cellular senescence in a p53-dependant manner in vitro. Additionally, we showed that ZNF768 interacts with and represses p53 phosphorylation and activity. Based on cancer genomic databases, we found that *ZNF768* overexpression is common in a variety of human cancers suggesting that elevated levels of ZNF768 could serve as a way to bypass oncogene-induced senescence and sustain tumor proliferation. We confirmed this finding in a small cohort of NSCLC where we observed elevated protein levels of ZNF768 in many tumors compared to matched lung tissue [13]. Altogether, these results highlight a new role for ZNF768 in repressing senescence and promoting cancer cell proliferation.

Despite evidence showing that ZNF768 expression is elevated in lung tumors, it is currently unknown whether these findings extend to a larger cohort and whether ZNF768 expression associates with clinicopathological features in LUAD. Here, using tissue microarrays, we report that high levels of ZNF768 is a common feature in these tumors. Furthermore, ZNF768 expression measured with the immunohistochemistry score (H-score) correlates with high proliferative clinicopathological features including the mitotic score and Ki-67 expression in the tumors. Further supporting the proliferative signature associated with ZNF768 in LUAD, knockdown of ZNF768 in different lung cancer cell lines severely impaired proliferation and decreased the expression of key cell cycle effectors. Altogether, our findings show that ZNF768 protein levels are elevated in LUAD, that ZNF768 associates with proliferative markers in this cancer and that this transcription factor is required to support cancer cell proliferation. Our study, that extends previous in vitro data, provides the first clinical observations supporting a possible role for ZNF768 in promoting cancer cell proliferation and tumor development in humans.

## 2. Materials and Methods

### 2.1. Patient and Tissue Selection, Tissue Microarray Construction

The study population included a selection of 577 patients that underwent a lung surgical resection for an adenocarcinoma between 2003 and 2012 at the Institut Universitaire de Cardiologie et de Pneumologie de Québec (IUCPQ). A consent form was obtained by the institution’s Biobank from all participants. Clinical and pathological data, including age, sex, smoking status, tumor stage (American Joint Committee on Cancer AJCC 8th edition) and overall survival, were retrieved using the electronic medical files and pathology reports. A thoracic pathologist (PJ) reviewed each tumor’s histologic slides and a representative area of the lesion was identified on one of them. The corresponding formalin-fixed, paraffin-embedded tissue blocks were retrieved, and tissue microarrays (TMA) were prepared by punching the donor block in the corresponding identified area on hematoxylin and eosin (H&E) slide. Tumor representative cores of 1.0 mm in diameter were placed 1.0 mm apart from each other on the recipient TMA paraffin block with a tissue arrayer (Beecher Instruments, Estigen, Estonia).

### 2.2. Immunohistochemistry

Five-micrometre-thick sections were cut from the TMA blocks on a microtome and placed on charged slides. For ZNF768 immunohistochemistry (IHC) (Aviva Systems Biology, San Diego, CA, USA, polyclonal, FLJ23436) slides underwent heat-induced epitope retrieval in a DAKO PT-Link using EnVision FLEX Target Retrieval Solution, Low pH (Citrate buffer pH 6.1), followed by an automatized IHC protocol on Dako Autostainer Link 48 (Dako, Agilent Technologies, Santa Clara, CA, USA). A previous validation set with positive tissues allowed to set ZNF768 antibody dilution to 1:100 with an incubation time of 20 min. The remaining of the IHC procedure was a standard Autostainer protocol using Dako EnVision FLEX+ kit reagents. After IHC, slides were counter-stained using EnVision FLEX hematoxylin. Ki-67 and p53 IHCs (respectively clones MIB1 and DO-7, Dako, Agilent Technologies, Santa Clara, CA, USA) were carried out on the same platform according to the manufacturer’s specific protocols.

### 2.3. Slide Digitalization and Pathologist’s Scoring of Protein Expression

Scoring of ZNF768, p53 and Ki-67 were done on digitalized slides. Digital images of each TMA stained with standard H&E and all IHCs at 20 × magnification were obtained using a slide scanner (NanoZoomer 2.0-HT; Hamamatsu, Bridgewater, NJ, USA). Whole slide images were visualized using the companion software (NDP view, Hamamatsu, Japan). Cores were excluded if they had fewer than a hundred tumor cells. ZNF768 and p53 expression were evaluated using the H-score method. This method combines the percentage of stained nuclei and the staining intensity using the following equation to obtain a score between 0 and 300: % nuclei of intensity 1 × 1 + % nuclei of intensity 2 × 2 + % nuclei of intensity 3 × 3. Ki-67 expression was evaluated using the percentage of stained nuclei on a scale of 0 to 100%.

### 2.4. Statistical Analysis

Associations with the ZNF768 *H*-score and background as well as clinical indicators were investigated using parametric (*t*-test; analysis of variance; Pearson *r*) or non-parametric (Spearman *rho*) analyses, depending on the nature of the variables and their distribution. All analyses were two-tailed with α = 0.05.

### 2.5. Cell Culture and Reagents

All cell lines (A549, NCI-H441 and NCI-H460, 293T) were obtained from American Type Culture Collection (ATCC, Manassas, VA, USA). A549 and 293T cells were cultured in complete Dulbecco’s Modified Eagle Medium (DMEM), while NCI-H441 and NCI-H460 cells were cultured in Roswell Park Memorial Institute (RPMI) 1640 medium. Both mediums were supplemented with Fetal Bovine Serum (FBS) (10%) (Wisent, #098-150, lot 185730, St-Bruno, QC, Canada) and penicillin-streptomycin (1%) (Wisent, #450-201-EL, St-Bruno, QC, Canada).

### 2.6. Virus Production and Transduction

Lentiviral shRNAs were obtained from the collection of The RNAi Consortium (TRC) at the Broad Institute (Cambridge, MA, USA). These shRNAs are named with the numbers found at the TRC public website: sh_Luciferase (TRCN0000072246), shZNF768_1 (TRCN0000017384), shZNF768_2 (TRCN0000017385). Lentiviruses were produced using psPAX2 and pMD2G as the packaging system. 293T cells were transfected with the vectors. Virus-containing supernatant was collected 48 h after transfection and filtered using a 0.45 µm filter. Cells were transduced for 24 h in the presence of 8 µg/mL polybrene. After the transduction, cells were dispersed into fresh medium. Cells were selected on the subsequent days using 2 µg/mL puromycin.

### 2.7. Western Blotting

All cells were rinsed twice with ice-cold PBS before lysis. Cells were lysed with Triton-X 100 containing lysis buffer (50 mM HEPES, pH 7.4, 2 mM EDTA, 10 mM sodium pyrophosphate, 10 mM sodium glycerophosphate, 40 mM NaCl, 50 mM NaF, 2 mM sodium orthovanadate, 1% Triton-X 100, and one tablet of EDTA-free protease inhibitors per 25 mL). Cells were rotated at 4 °C for 10 min and then the soluble fractions of cell lysates were isolated by centrifugation for 10 min in a microcentrifuge. Protein levels were then quantified using Bradford reagent and analyzed by Western blotting. Protein extracts were diluted in sample buffer, denatured by heat (95 °C) for 10 min and loaded on precast gels (Life Technologies, Carlsbad, CA, USA). Proteins were transferred to PVDF membranes blocked in 5% milk diluted in PBS-Tween and incubated with their primary antibody overnight at 4 °C. The following antibodies were used: ZNF768 (Aviva Systems Biology, San Diego, CA, USA, FLJ23436, dilution 1:1000); anti–β-actin (Cell Signaling Technology, Danvers, MA, USA, #4967, dilution 1:1500). Secondary antibodies were purchased for Cell signaling technology (#7074S, dilution 1:5000). Amersham ECL Western Blotting Detection Reagent (RPN2106) was used to image the blots.

## 3. Results

### 3.1. Analysis of ZNF768 in LUAD Tissue Microarrays

To characterize in detail the expression of ZNF768 in a large cohort of LUAD, immunohistochemistry (IHC) was performed on a tissue microarray (TMA) constructed using LUAD collected from 364 patients that underwent a wedge resection (*n* = 18), segmentectomy (*n* = 37), lobectomy (*n* = 270), pneumonectomy (*n* = 24), bilobectomy (*n* = 23) or another type of surgical resection (*n* = 2). A detailed summary of the cohort’s characteristics is presented in Table 1.

ZNF768 is a protein that contains C2H2 domains in its C-terminal section and localizes to the nucleus in cancer cell lines. To validate the specificity of the ZNF768 antibody, an IHC was first conducted on tissues from wild-type mouse and ZNF768 complete knockout mouse as positive and negative controls respectively (Appendix A). The absence of signal in all ZNF768 knockout tissues compared to wild-type mice confirmed the specificity of the staining. Here, we confirmed that ZNF768 staining in the tumors was predominantly nuclear, as seen in vitro (Figure 1A). Interestingly, in addition to the nuclear staining, we also observed some cytoplasmic staining in a small number of cases. Although this cytoplasmic staining was not observed often, it was usually of high intensity and of granular appearance.

To compare the different levels of ZNF768 expression in the tumors, the H-score method was used, which combines the percentage of the positively stained cells with the intensity of the staining to give a semi-quantitative score between 0 and 300. The average H-score obtained for ZNF768 was 130 in this cohort. To classify the intensity of the ZNF768 IHC staining, a three-pronged system was used based on the nuclear staining only. We found that the average ZNF768 staining intensity varied substantially between the tumors (Figure 1B,C). More specifically, tumors with a score of 1, meaning lower ZNF768, represented about half of our cohort with 53%, while tumors with scores of 2 or 3 represented the other half with 38.5% and 8.5%, respectively. When looking at the percentage of ZNF768 IHC positively stained cells, we found that it had a highly negatively skewed distribution meaning that the majority of the values are concentrated to the right of the distribution (Figure 1D). With the median being 90 in our cohort, this result highlights the fact that most tumor cells in LUAD express ZNF768. Consistent with our previous findings, these results show that ZNF768 is heterogeneously expressed in LUAD and that high levels of this protein are a common feature of these tumors.

### 3.2. ZNF768 Expression Associates with Ki-67 and Proliferative Clinicopathological Features in LUAD

Despite having observed high levels of ZNF768 in LUAD and in certain types of cancer, the exact functions of ZNF768 in cancer remain unknown. To better define the effect of high levels of ZNF768 in tumors and to verify its clinical relevance, we examined the pairwise correlations between the ZNF768 H-score with different clinicopathological characteristics in our cohort. The results of these analyses are presented in Table 2. Consistent with studies showing that ZNF768 promotes cell proliferation and survival in vitro, we observed a positive correlation between ZNF768 levels and the well-known proliferation marker Ki-67 (Figure 2A–C and Appendix A). A positive correlation between ZNF768 and the mitotic score of the tumors was also noted, further supporting the increased proliferation associated with high levels of ZNF768 (Figure 2D). We next looked at the other clinicopathological features associated with ZNF768 and found an association with sex, pleural invasion and with certain histologic patterns of the tumors (Figure 2E,F). For instance, we found that the highest levels of ZNF768 were observed in solid, followed by acinar pattern. In contrast, papillary pattern showed the lowest levels of ZNF768, while lepidic and micropapillary pattern showed intermediate levels of ZNF768. Of note, we did not find any association between ZNF768 and overall survival (Hazard ratio = 1.0, ns). However, ZNF768 possibly associates with tumor stage in LUAD with a *p*-value just above the alpha in our cohort and higher levels of ZNF768 in tumors of stage 3 and above. Given that pleural invasion and some histologic patterns are generally associated with worse prognosis, we next asked whether high levels of ZNF768 in these categories were also associated with higher proliferation. To do so, we compared ZNF768 levels and the percentage of Ki-67 positive cells. Interestingly, we observed similar patterns of expression for ZNF768 and Ki-67 in the different categories (Figure 2G,H). Altogether, these results indicate that high ZNF768 levels associate with proliferation in LUAD.

### 3.3. Depletion of ZNF768 Impairs Proliferation in Lung Cancer Cell Lines

Given the correlation between ZNF768 levels and proliferative clinicopathological features in LUAD, we investigated whether repressing ZNF768 could impact proliferation of lung cancer cells in vitro. ZNF768 was knockdown in various NSCLC cell lines and proliferation was assessed. Three NSCLC cell lines with different mutation signatures were used, i.e., two adenocarcinoma cell lines (A549 and NCI-H441) and one large cell carcinoma cell line (NCI-H460). All three cell lines are mutated for the K-RAS oncogene, while both the A549 and NCI-H460 cell lines contain mutations activating the PI3K/mTOR pathway (A549: *STK11*, NCI-H460: STK11-PIK3CA). Lastly, of the three cell lines, only the NCI-H441 is p53 mutant. We first identified two short hairpin RNA (shRNA) that effectively knockdown ZNF768 (Figure 3A–C and Appendix A). Using these hairpins, we observed that ZNF768 knockdown greatly reduces proliferation in all these cell lines. No differences between the two hairpins were noted. This result agrees with previous findings where we showed that ZNF768 controls proliferation in different cancer cell lines [13]. Collectively, these results highlight a direct effect of ZNF768 on proliferation in lung cancer cell lines and further support a link between ZNF768 and proliferative clinicopathological features in LUAD.

### 3.4. ZNF768 Depletion Alters the Expression of Key Genes Regulating Proliferation in Lung Cancer Cells

ZNF768 is a transcription factor that has been shown to regulate a wide variety of genes in a cell-specific manner by binding to mammalian-wide interspersed repeats (MIRs) in vitro [14]. It was recently shown that the transcriptional signature associated with ZNF768 depletion was enriched in pathways regulating cell cycle progression [13]. Based on the impact of ZNF768 depletion on lung cancer cells, we aimed to determine whether ZNF768 depletion also altered the expression of genes involved in cell cycle progression. To do so, ZNF768 was knockdown in A549, NCI-H441 and NCI-H460 cells and RNA was extracted following selection. Several genes controlling the cell cycle (e.g., *CDK1*, *MYBL2*, *CCNB1*, *CCNB2*), chromosome segregation (e.g., *SMC4*, *NUSAP1*, *AURKA*, *AURKB*, *PLK1*, *CDC20*), genome replication (e.g., *PCNA*, *CDC45*, *TOP2A*) and stability (e.g., *BRCA1, DDB2, EZH2, MYBL2, TOP2A*) were next measured. These genes were selected based on previous transcriptomics assays performed following the knockdown of ZNF768 in U87 cells [13] or the overexpression of a dominant negative ZNF768 in U2OS cells [14]. Some of these genes were also shown to be regulated following ZNF768 depletion in several cell lines in the iLincs resource, a publicly available databank providing the expression profile of almost 1000 genes (L1000 assay) in response to various perturbagens in multiple cell lines [15]. As shown in Figure 4, we found that ZNF768 depletion was associated with a severe decrease in the expression proliferative genes (Figure 4A–C). Although the global effect on gene expression was consistent, the extent of repression was slightly different in the three lines, with the highest repression in NCI-H441 cells and the lowest repression in NCI-H460 cells. Overall, these results support a strong connection between ZNF768 expression and proliferation in lung cancer cell lines.

## 4. Discussion

ZNF768 is a newly characterized transcription factor that is required for cell proliferation [13,14]. We recently showed that ZNF768 is phosphorylated and destabilized in response to RAS pathway activation [13]. We found that this transcription factor regulates a core set of proliferative genes but also interacts with and represses p53 phosphorylation and activation in vitro. Furthermore, we measured high ZNF768 protein levels in a small cohort of NSCLC compared to healthy lung tissue suggesting a possible role of ZNF768 in tumorigenesis. Despite evidence that ZNF768 is overexpressed in NSCLC, the exact role of ZNF768 in cancer and its clinicopathological relevance remain unknown. Here, we report that high levels of ZNF768 are frequently observed in LUAD and that these levels positively correlate with high proliferative clinicopathological features of the tumors. These observations, that support previous in vitro findings, provides the first clinical observations of a possible role for ZNF768 in supporting cancer cell proliferation and tumor development in humans. Along with these results, we also show that ZNF768 depletion directly impairs proliferation of different NSCLC cell lines. This effect of ZNF768 knockdown was associated with a severe repression in the expression of several genes controlling the cell cycle. Our results support a central role for ZNF768 in promoting the proliferation of lung cancer cells.

ZNF768 protein levels are elevated in LUAD. Our findings, that complement preliminary observations made in a smaller cohort of patients, also revealed a link between ZNF768 and proliferative markers in LUAD. As recently discussed, preliminary analyses through the Human Protein Atlas resource showed that several cancers display high ZNF768 protein expression [13]. Autoantibodies against ZNF768 are also detected in the plasma of patients with colorectal cancer [16,17]. These results suggest that other cancer cell types may exploit ZNF768 to promote proliferation. An interesting question that emerges from these observations relates to the mechanisms driving ZNF768 expression cancer cells. Although we found a severe rise in ZNF768 proteins levels in LUAD compared to normal lungs, this effect was not linked to profound changes in *ZNF768* mRNA transcription [13]. Because *ZNF768* gene is rarely amplified in these tumors, these results suggest that post-transcriptional and/or post-translational processes probably take place to promote ZNF768 protein levels in cancer. Overall, these observations indicate that a disconnection between *ZNF768* mRNA expression and ZNF768 protein levels exists in LUAD, and that measuring ZNF768 protein levels is key to study this transcription factor in this cancer.

ZNF768 expression correlates with high proliferative clinicopathological features in LUAD. In cancer cell lines, ZNF768 promotes the expression of key genes required for proliferation [13,14]. Supporting the established role of ZNF768 in modulating gene expression, we found that ZNF768 staining in the tumors was predominantly nuclear. We also observed cytoplasmic ZNF768 staining in a small number of LUAD. Although this cytoplasmic staining was not observed often, it was usually of high intensity and of granular appearance. This observation appears to be specific to tumor cells in vivo as ZNF768 was never observed outside the nuclei in cancer cell lines in vitro [13,14]. The biological processes that contribute to ZNF768 cytoplasmic accumulation in LUAD as well as the role of ZNF768 in this cell compartment remain to be determined.

There is growing interest in the study of the role of transcriptional regulators in cancer. Transcription factors, cofactors and chromatin regulators can contribute to tumorigenesis by altering the core transcriptional regulatory circuits of the cells to drive proliferation and survival [18,19,20]. Here, we provide data suggesting that ZNF768 might be one important transcription factor promoting proliferation in LUAD. The striking loss of proliferative capability and the reduction in the expression of proliferative genes in lung cancer cells depleted from ZNF768 support this conclusion. It should be noted that the impact of ZNF768 deletion on proliferative gene expression was not the same in all three cell lines tested, with the highest repression in NCI-H441 and the lowest repression in NCI-H460. The reason for these differences is unknown. It is possible that differences in the basal expression of these targets may have affected the impact of ZNF768 depletion on their expression. Additionally, ZNF768 was previously shown to control gene expression in a cell-specific manner by regulating the expression of an array of other transcription factors [14]. This could have contributed to generate differences in gene expression between the cell lines depleted from ZNF768. Lastly, we need to point out that basal ZNF768 protein levels were different between the cell lines, with NCI-H441 showing the higher levels compared to A549 and NCI-H460. Interestingly, these cells are the ones that were the most affected by ZNF768 depletion in both the repression of cell cycle genes and the proliferation rate, suggesting that this cell line may be more ‘addicted’ to high ZNF768 expression. This could also explain why the impact of ZNF768 depletion on the proliferative gene signature was different between the cell lines.

We have previously reported that, in addition to control the expression of proliferative genes, ZNF768 also supports cell proliferation by directly repressing p53 [13]. In vitro, ZNF768 physically interacts with p53 to repress its phosphorylation and prevent its full activation [13]. These observations suggest that ZNF768 might also promote LUAD development by repressing the tumor suppressive functions of p53. Over the years, several negative regulators of p53 were shown to be amplified and/or overexpressed in cancer [21,22,23,24,25,26,27]. The best example is the E3-ubiquitin ligase mouse double minute 2 (MDM2) that triggers the proteasome-dependent degradation of p53. Hyperactivation of MDM2 is found in many cancers where it represses p53 [22,23,24,25,27]. Although ZNF768 unlikely modulates p53 to the same extent as MDM2, our observations support the possibility that ZNF768 overexpression might fulfill similar functions in the development of LUAD.

ZNF768 expression is linked to proliferation in LUAD but is not a predictor for the overall survival of the patients in our cohort. These results suggest that ZNF768 is not a suitable prognostic factor in LUAD. Although this finding may look surprising, it is important to point out that tumour cell proliferation and survival are not always associated in cancer. For example, the link between the proliferation marker Ki-67 and survival is still controversial in some types of cancer and has limitations depending on the study [28,29,30]. Furthermore, while mutations in driver oncogenes such as K-RAS in lung cancer are commonly associated with poor outcome [31,32,33], many studies failed to detect a significant association between these clinical data and prognosis [34,35,36]. Hence, the lack of an association between ZNF768 and prognosis in LUAD does not detract from its association with proliferation in this cancer and its possible contribution to tumorigenesis.

Our study suggests a possible role for ZNF768 in LUAD as a pro-proliferative transcription factor. Nevertheless, many questions remain concerning the exact impacts of ZNF768 in LUAD and its oncogenic potential. Foremost, the genomic context in which ZNF768 protein levels are elevated in LUAD is still unknown. Secondly, while in vitro evidence and cancer proteomics point to a possible role of ZNF768 in LUAD, additional studies are needed to define the exact contribution of ZNF768 in tumorigenesis. The use of animal models of cancer such as LUAD driven by oncogene(s) or xenografts with altered levels of ZNF768 could establish whether ZNF768 directly contributes to tumor formation in vivo. In parallel, it would be interesting to determine whether ZNF768 overexpression alone has oncogenic capabilities in vivo. Testing these hypotheses could help define the exact functions of ZNF768 in cancer and explore its therapeutic potential in this disease.

## 5. Conclusions

In conclusion, we report that ZNF768 levels are commonly elevated and correlate with high proliferative markers in LUAD. We provide evidence that ZNF768 knockdown greatly impairs proliferation in NSCLC cell lines, which is accompanied by decreased expression of numerous genes involved in cell cycle progression and genomic integrity. These findings indicate that high levels of ZNF768 could offer a proliferative advantage to lung cancer cells and thus, might contribute to tumor formation. Further studies are required to establish the direct role of ZNF768 in human cancer and its possible mechanisms of action in vivo.

## Figures and Tables

**Figure 1 cancers-13-04136-f001:**
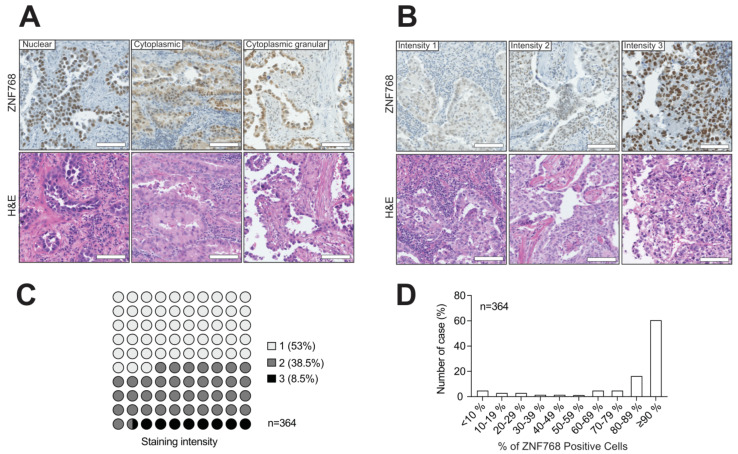
ZNF768 expression is often elevated in human LUAD. (**A**) Representative pictures of the different ZNF768 immunohistochemistry staining patterns in the tumors (nuclear–cytoplasmic-cytoplasmic granular) (magnification × 20, scale bar 100 µm). (**B**) Example of the three-pronged scoring system (intensity 1-intensity 2-intensity 3) (magnification × 20, scale bar 100 µm). (**C**,**D**) Distribution of the main staining intensity and the percentage of ZNF768 positive cells in the tumors of the cohort.

**Figure 2 cancers-13-04136-f002:**
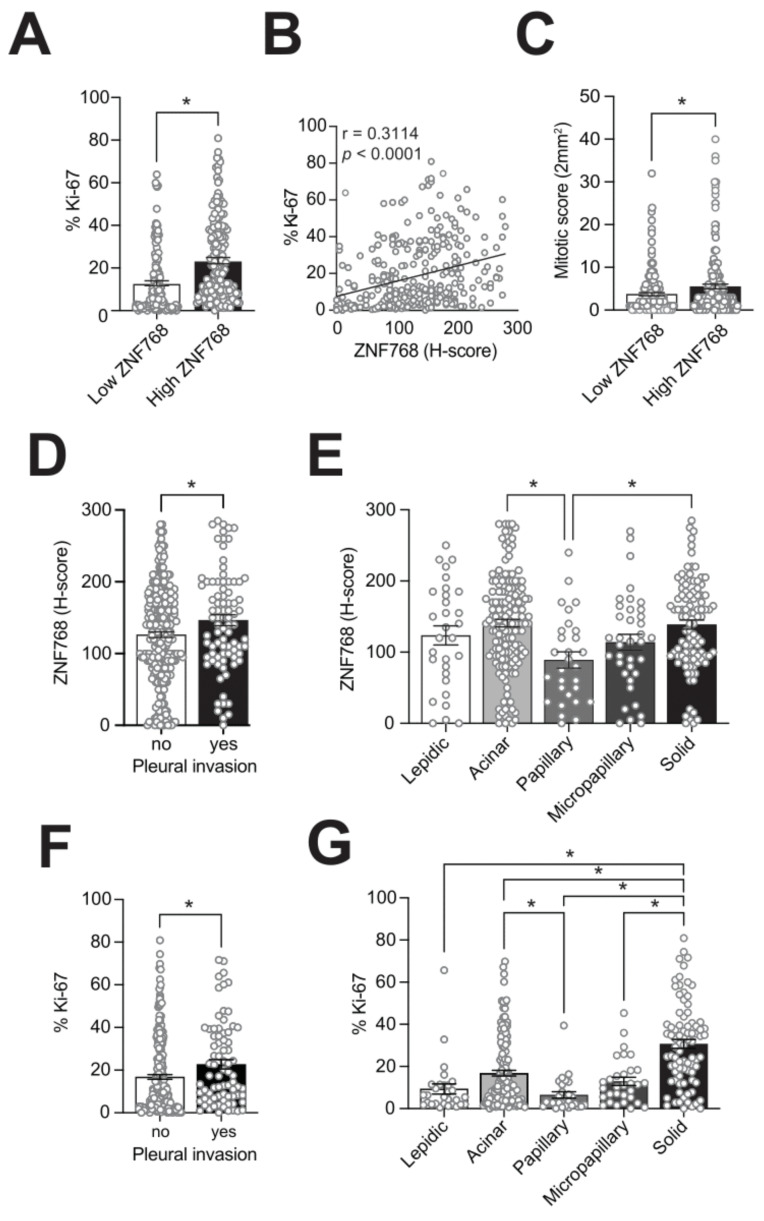
ZNF768 expression associates with proliferative clinicopathological features in LUAD. (**A**) Tumors were separated in high and low ZNF768 expressing groups based on the H-score of the tumor (low, ZNF768 H-score ≤ 130; high, ZNF768 H-score > 130). Difference in the Ki-67 percentage between high and low ZNF768 expressing groups. (**B**) Correlation between the ZNF768 H-score and the percentage of Ki-67 positive cells. (**C**) Difference in the mitotic score between high and low ZNF768 expressing groups. (**D**) Differential expression of ZNF768 in the tumors with pleural invasion. (**E**) ZNF768 expression in the different histological patterns of the tumors. (**F**) Differential percentage of Ki-67 positive cells in the tumors with pleural invasion. (**G**) Ki-67 percentage in the different histological patterns of the tumors. Significance was determined by Spearman’s correlation, *t*-tests (2 variables) or ANOVAs (< 2 variables) depending on the nature of the factors (* *p* < 0.05).

**Figure 3 cancers-13-04136-f003:**
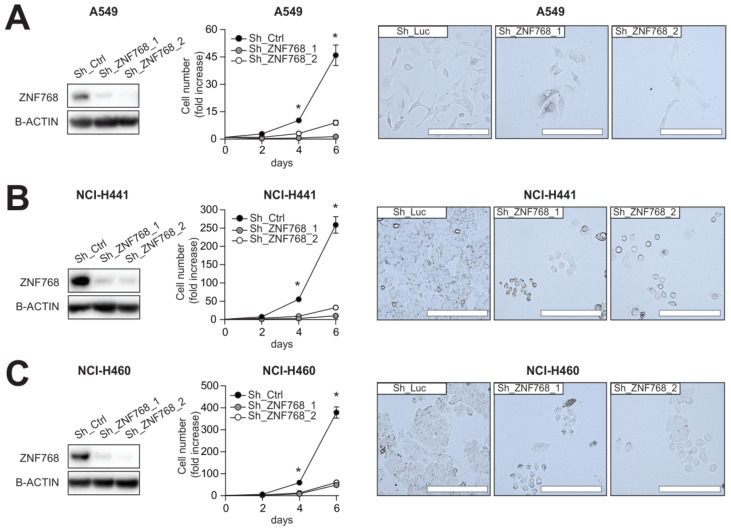
ZNF768 depletion impairs proliferation of various lung cancer cell lines. (**A**) A549 cells, (**B**) NCI-H441 and (**C**) NCI-H460 cells were transduced with lentivirus expressing shRNA targeting ZNF768. Cells were selected with puromycin and protein were extracted, or cells were plated and counted every two days. Pictures were taken on the last day of counting. Representative pictures are shown. Western blot analyses were performed to confirm ZNF768 depletion. This experiment was performed at least 2 times for each cell types and cell counts were done in triplicates. Significance was determined by Two-way ANOVA (* *p* < 0.05 versus control).

**Figure 4 cancers-13-04136-f004:**
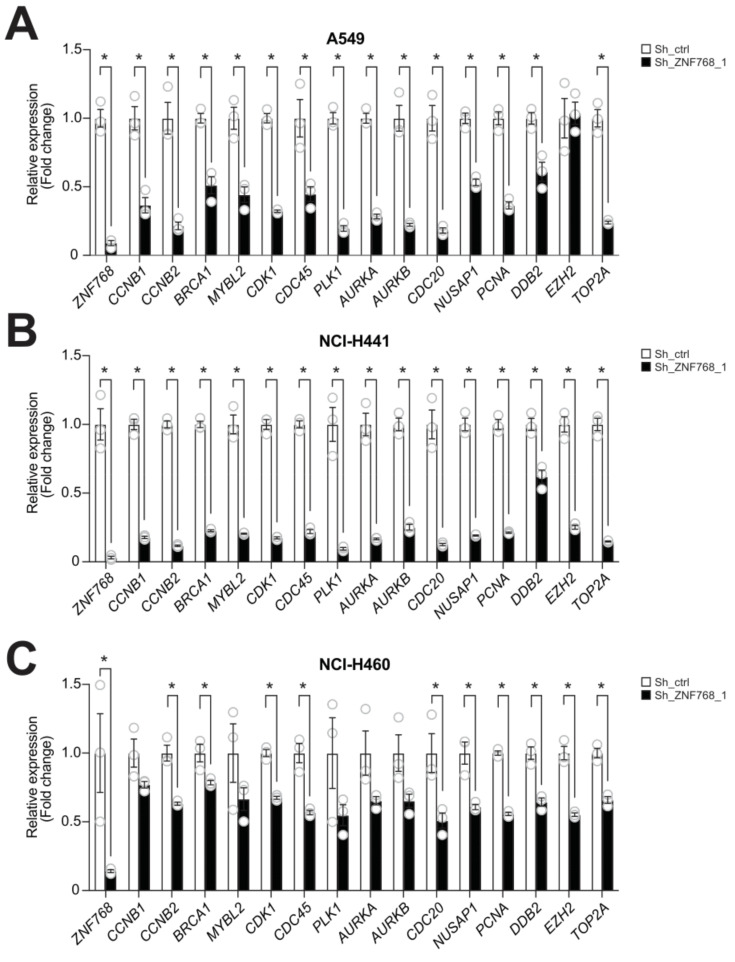
ZNF768 depletion alters the expression of key genes regulating proliferation in lung cancer cells. (**A**) A549 cells, (**B**) NCI-H441 and (**C**) NCI-H460 cells were transduced with lentivirus expressing shRNA targeting ZNF768 and were selected with puromycin. RNA was harvested from cells following selection and the expression of proliferative genes was measured by qPCR (*n* = 3/condition). This experiment was performed once. Significance was determined by *t*-test (* *p* < 0.05 versus control).

**Table 1 cancers-13-04136-t001:** Clinical characteristics of the cohort.

Characteristics	No. (*n* = 364)	%
**Age**, M(SD) [min–max]	64 (8.8) [33–83]	
**Sex**		
Male	152	41.8
Female	212	58.2
**Smoking status**		
Nb. pack year, M(SD) [min–max]	41 (23.8) [0–156]	
Non-smoker	30	8.2
Former	237	65.1
Active	89	24.5
Passive	8	2.2
**Stage (AJCC 8th edition)**		
I	238	65.4
II	71	19.5
III	48	13.2
IV	7	1.9
**Follow-up**		
Months, M(SD) [min–max]	94.3 (36.9) [3.3–197.7]	
**Survival**		
5-year survival	281	77.2

Abbreviations: M, mean; SD, standard deviation; Min, minimum; Max, maximum; Nb, number.

**Table 2 cancers-13-04136-t002:** Associations between ZNF768 *H*-score and clinicopathological features in LUAD using univariate analyses.

Variable	Value	*t*-Test	ANOVA	Correlation	*p*
		t	F	r/rho	
Sex	Male, female	−2.12			0.03
Age	Continuous variable			0.04	ns
Smoking status	Non-smoker, former, active, passive		0.82		ns
Pack/year	Continuous variable			−0.01	ns
Max. tumor size	Continuous variable			−0.06	ns
Histological pattern	Lepidic, acinar, papillary, micropapillary, solid		4.58 (Acinary, Solid > Papillary)		<0.001
Lymphovascular invasion	Yes, no	−1.00			ns
Necrosis	Yes, no	−1.28			ns
Pleural invasion	Yes, no	−2.36			0.02
Cytological grade	Low, high	−1.11			ns
Spread through air spaces (STAS)	Yes, no	0.30			ns
Distance of STAS from tumor	Continuous variable			−0.06	ns
Inflammation	None, Acute, chronic, acute and chronic		0.27		ns
Pathological stage (AJCC 8th ed.)	1, 2, 3&4		2.63		0.07
Mitotic score (2 mm^2^)	Continuous variable			0.12	0.03
Ki-67 (%)	Continuous variable			0.31	<0.0001
Progression	Yes, no	−1.03			ns

ns: non-significant.

## Data Availability

No new data were created or analyzed in this study. Data sharing is not applicable to this article.

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
