# Peer review of "ZNF768 Expression Associates with High Proliferative Clinicopathological Features in Lung Adenocarcinoma"

_cancers, 2021, doi:10.3390/cancers13164136_

Round 1

Reviewer 1 Report

The authors in their study of ZNF768 expression in lung adenocarcinoma have provided some interesting findings for the role of ZNF768 as a proliferative marker in LUAD.

However, the study has a few shortcomings.

  1. In Figure 1 of the Results section, the authors report expression of ZNF768 in 53% of LUAD with score of 1 (low ZNF768 expression) compared to 38.5 (expression score 2) and 8.5% (expression score 3). The authors do not specify whether the expression score is linked with tumor grade hence they observed this pattern of expression.
  1. Similarly, in the Results 3.2, the authors speculate that “ZNF768 possibly associates with tumour stage in LUAD with a p-value just above the alpha in our cohort study and higher levels of ZNF768 in tumors of stage 3 and above”. It is not evident whether the authors did perform statistics for ZNF768 expression levels with tumour grade to support their assumptions.

In both cases, the authors need to examine whether an association with ZNF768 expression levels and tumor grade exists before drawing any conclusions.

3. The findings from knockdown and depletion experiments were performed in NSCLC established cell lines. Several studies have shown that the tissue culture environment influences the expression of genes and is associated with the loss of tumorigenic phenotype. It would be interested to investigate whether similar results could be observed in primary LUAD cell cultures. Have the authors tested any of the biomarkers for LUAD in their cell lines prior to their expreriments? eg. EGFR mutation?

  1. In the Results 3.4, the authors report altered expression levels for a number of key genes regulating cell proliferation in lung cancer. How the authors decided to use these genes? Is the altered gene expression the result of ZNF768 depletion or the result of the effect of gene mutations already in the established cell lines that they have used in their experiments?

Reviewer 2 Report

The manuscript by Poirier et al investigated the role of ZNF768 in mediating  proliferation of cancer cells in lung adenocarcinoma. Importantly, they also show that ZNF768 expression associates with clinicopathological features of lung adenocarcinoma.

Authors should address the following,

1. It is necessary to clearly state how this study and its findings advance the scientific knowledge compared to the reference 13. It seems that ZNF768 is already know to be over expressed in cancers. 

2. Publicly available data bases such as TCGA or other available data should be analyzed and presented the results to broaden the scope. Additionally, authors could also investigate the alteration of expression upon different stages or progression of LAUD.

3. Since ZNF768 regulates senescence, authors should perform IHC on TMA with senescence marker.

4.  Authors should also perform xenograft studies, including analyzing Ki67, cell death and senescence markers, to demonstrate the in vivo effects of ZNF768 depletion.

Reviewer 3 Report

The manuscript submitted by Poitier et al. addresses the role of a zinc-finger transcription factor (ZNF768) in proliferation in lung carcinoma. Although the text is clear and well written, a few aspects could be improved before its publication. Additionally, images supporting some of the conclusions are needed. A detailed list can be found below:

  1. In the results section, the significant correlations shown in Table 2 must be explained.
  2. The main finding that allows the authors to associate the expression of ZNF768 with the proliferation of lung adenocarcinoma in vivo, is the correlation with Ki-67. To support the data presented in Figure 2, representative images must be shown as a separate figure or additional panels in the figure.
  3. The authors also analyzed the effect of the depletion of ZNF768 in the genes regulating proliferation in three lung cancer cell lines. Although the global effect on proliferation was consistent, it should be commented that the extent of repression was different in the three lines. In NCI-H460, the decrement of some genes (i.e, AURKA and AURKB) was not significant. This fact must be described in the text. Does the extent of repression correlate with the effects on the proliferation rate?
  4. The possible causes of the presence of cytoplasmic granules in some samples must be commented on in the discussion.

Minor details:

- In the abstract, numerous lung cancer cell lines must be replaced with several lung cancer cell lines.

- Add a reference to the results described in first paragraph of the discussion.

Round 2

Reviewer 1 Report

Dear authors,

Thank you very much for addressing my comments. 

Author Response

We want to thank referee #1 for the review of our manuscript. 

Reviewer 2 Report

To confirm novelty of the study, the authors have not provided their approved Nature communications manuscript (reference 13), which is not publicly available yet.

In response to comment 2, authors provided figures from their nature Communications paper. No more additional figures are provided to this particular manuscript. 

In response to comment three, it is definitely possible to perform senescence markers in bigger cohorts. Since this would broaden the scope/impact of the study.

In response to comment 4, authors feel that doing xenograft may not add any more info to the paper, which I feel partially true, however,  it would give credibility to their observations.

Round 3

Reviewer 2 Report

Thank you for providing the manuscript.

Recommend to accept the paper in current form.